# Leydig Cell Tumors of the Testis: An Update of the Imaging Characteristics of a Not So Rare Lesion

**DOI:** 10.3390/cancers14153652

**Published:** 2022-07-27

**Authors:** Florian Maxwell, Alexia Savignac, Omar Bekdache, Sandra Calvez, Cédric Lebacle, Emmanuel Arama, Nada Garrouche, Laurence Rocher

**Affiliations:** 1Department of Radiology, Bicêtre Hospital, 63 Rue Gabriel Péri, 94270 Le Kremlin Bicêtre, France; alexia.savignac@aphp.fr (A.S.); omar.bekdache@aphp.fr (O.B.); sandra.calvez@aphp.fr (S.C.); 2Department of Urology, Bicêtre Hospital, 63 Rue Gabriel Péri, 94270 Le Kremlin Bicêtre, France; cedric.lebacle@aphp.fr; 3Department of Radiology, Antoine-Béclère Hospital, 157 Rue de la Porte de Trivaux, 92140 Clamart, France; emmanuel.arama@aphp.fr (E.A.); nada.garrouche@aphp.fr (N.G.)

**Keywords:** testicular stromal tumors, Leydig cell tumors (LCT), Klinefelter’s syndrome (KS), Leydig cell hyperplasia, ultrasound (US), Doppler, contrast-enhanced ultrasound (CEUS), elastography, magnetic resonance imaging (MRI)

## Abstract

**Simple Summary:**

Stromal tumors of the testis are rare. However, among this group, Leydig cell tumors (LCT) are the most frequent, and recent studies suggest that LCTs account for up to 22% of small testicular nodules. It is now accepted that small LCTs can benefit from testis-sparing surgery or in some selected cases radiological surveillance. Since percutaneous testicular biopsy is still not recommended, the diagnosis of LCT rests on multimodal imaging techniques. Therefore, it is essential for the radiologist and the urologist to know the main imaging features of LCTs in ultrasound and MRI.

**Abstract:**

Pre-operative testicular tumor characterization is a challenge for radiologists and urologists. New data concerning imaging approaches or immunochemistry markers improve the management of patients presenting with a testicular tumor, sometimes avoiding radical orchiectomy. In the past 20 years, imaging modalities, especially ultrasound (US) and magnetic resonance imaging (MRI), improved, allowing for great progress in lesion characterization. Leydig cell tumors (LCT) are rare testicular tumors developing from the stromal tissue, with relatively scarce literature, as most of the studies focus on the much more frequent germ cell tumors. However, with the increase in testicular sonography numbers, the incidence of LCT appears much higher than expected, with some studies reporting up to 22% of small testicular nodules. Multimodal ultrasound using Doppler, Elastography, or injection of contrast media can provide crucial arguments to differentiate LCT from germ cell tumors. Multiparametric MRI is a second intention exam, but it allows for extraction of quantifiable data to assess the diagnosis of LCT. The aims of this article are to review the latest data regarding LCT imaging features, using multimodal ultrasound and multiparametric MRI, and to focus on the peculiar aspect of the testis of patients with Klinefelter’s syndrome. The possibility of an LCT should be raised in front of a small hypoechoic tumor with a marked corbelling hypervascularization in an otherwise normal testicular pulp. Ultrasonographic modules, such as ultrasensitive Doppler, contrast-enhanced ultrasonography, or elastography, can be used to reinforce the suspicion of LCT. MRI provides objective data regarding vascularization and enhancement kinetics.

## 1. Introduction

Testicular tumors are one of the most frequent cancers in the young. A total of 95% of these lesions derive from the germ cell line with seminomatous and non seminomatous tumors. Sex cord stromal tumors are rare, as they represent only 5% of all testicular tumors. This group is dominated by one subtype of tumor: the Leydig cell tumor (LCT), as Sertoli cell tumors are extremely rare (<1% of all testicular malignancy) [1]. With the increase in the number of scrotal ultrasounds, notably in infertile patients, the discovery of non-palpable testicular nodules has become a frequent problem. Recent studies suggest that LCTs account for up to 22% of testicular nodules < 1.5 cm detected incidentally in infertile patients [2]. Leydig cells are in the testicular stromal tissue, and they are responsible for most of the testosterone secretion in men. The Leydig cells also have a paracrine function in supporting Sertoli cells and spermatogenesis. LCT have two frequency peaks: in the pediatric population (5–10 years) and in men mostly after the third decade. The tumorigenesis of LCT remains largely unknown, however a disruption of the hypothalamic pituitary axis is the preferred hypothesis, and notably an anomaly of the luteinizing hormone (LH) circulating level. Some predisposing factors have been identified as Klinefelter’s syndrome (47 XXY caryotype) or germline fumarate hydratase mutation. Historically, LCT were associated with gynecomastia and symptoms related to low testosterone level, but nowadays most of the lesions are asymptomatic. Pre therapeutic testicular marker assay (HCG, alpha fetoprotein, LDH) is mandatory to rule out some non seminomatous germ cell tumors; although, in the case of LCT, no specific biomarker is available. The macroscopic aspect of LCT is a typical golden-brown color, often confirmed in the hypothesis of an LCT during surgery. The stromal origin is confirmed using immunochemistry, as 100% of LCT express inhibin A. LCT are differentiated from Sertoli cell tumors using Calretinin A (strong cytoplasmic and nuclear expression in LCT, weak and exclusively cytoplasmic in Sertoli cell tumors). The malignant potential of LCT has been controversial in the literature, as most studies focused on stromal tumors (LCTs and Sertoli cell tumors). As it is widely accepted that LCT are always benign in the pediatric population, the malignant potential of LCT increases with age, with a peak around 60 years [3,4,5]. The evaluation of the malignant potential of LCT is based on the anatomopathological criteria of Kim extended by Albers in 2011 (size > 5 cm, cytological atypia, increased mitotic activity, increased MIB-1 expression, necrosis, vascular invasion, infiltrative margins, extra testicular extension, DNA aneuploidy) [1,6]. The historically accepted rate of 10% malignancy has been revised downwards from the 2000s thanks to progress in andrology and imaging that allow for an early (most often fortuitous) diagnosis of LCT. The most recent studies find a malignancy rate of 2.5% [5]. As most LCTs are considered benign in recent literature testis-sparing surgery is now the standard of care for a suspected LCT. Some selected young (<50 years old) patients with <5 mm lesions can also benefit from clinico-radiological surveillance [7,8]. Multiparametric techniques of scrotal imaging—ultrasound or magnetic resonance imaging (MRI)—improve diagnostic performance for LCTs and offer some patients therapeutic alternatives to radical orchiectomy, thus emphasizing the diagnostic challenge for the radiologist [8]. Klinefelter syndrome is a known risk factor for LCTs [9]. Our aim in this literature review is to present the imaging appearance of sporadic LCTs, as well as the testicular anomalies detectable in Klinefelter syndrome patients.

## 2. Literature Review Protocol and Eligibility Criteria

The literature search was performed by two senior radiologists specialized in testicular imaging, using the PubMed, EMBASE, and Cochrane databases, and covering articles from their first availability until December 2021. We considered different methods of ultrasound imaging and MRI and their variations (ultrasound, ultrasonography, sonography, US, B-mode ultrasound, grayscale ultrasound, Doppler, color Doppler, CD, CDUS, ultrasensitive Doppler, SMI^®^, CEUS, contrast-enhanced ultrasound/sonography, elastography, SWE, strain elastography, shear wave elastography, MRI, MR, multiparametric MRI/MR, contrast-enhanced MRI/MR, and diffusion-weighted/DWI MRI/MR), individually associated with the terms testicle/testis and Leydig/Leydigoma. The two senior radiologists reviewed titles and abstracts one by one, excluding posters, comments, and letters to the editor, duplicate articles, pediatric studies, and animal studies. Several articles in a language other than English were included because of their importance to the issue in question.

## 3. Multiparametric Ultrasound

### 3.1. B-Mode Ultrasound

A PubMed search for articles between 2011 and 2021 using the key words Leydig cell tumour/tumor—ultrasound/sonography yielded 75 references, 28 of which were case reports. B-mode ultrasound is the most studied imaging modality in the literature, as it is the reference for examination of the scrotum. In 2015, we reported the characteristics of 38 LCT:100% were solid and homogeneous, 68.4% of which were infracentimetric (median size 7.0 mm) [10]. The most voluminous LCTs were round or lobulated and most were hypoechogenic compared with the adjacent clearly demarcated pulp (Figure 1).

Analysis of the adjacent testicular pulp is primordial because most LCTs are found in normal pulp, and only 9.4% of the lesions were accompanied by non-grouped grade I (Richenberg et al.) microlithiasis [11]. These results accord with those reported by Maizlin et al. and Lock et al. [12,13]. The association of a testicular nodule with other focal alterations of the pulp and grouped microliths is highly suggestive of germ cell tumors, in particular seminoma, and should not in the first instance be taken as indicative of LCT [12,13,14].

### 3.2. Color Doppler and Ultrasensitive Doppler

The PubMed search for articles published between 1993 and 2021, using the key words Leydig cell tumour/tumor—ultrasound/sonography—SMI^r^—ultrasensitive Doppler yielded 24 references, including 6 case reports.

Color Doppler allows for qualitative assessment of tumor vasculature and its organization. Most recent ultrasound devices have a Doppler module optimized for the detection of the microflow associated with a decrease in motion artifacts (SMI^®^ from Canon or Angioplus RT^®^ from Aixplorer). Maxwell et al. found that close to 95% of lesions were hypervascular compared with the adjacent pulp, with an enveloping vascularization pattern or a mixed (central and peripheral) pattern of the vascular architecture [10]. These results corroborate the literature data [12] and the most recent case reports [15]. The presence of internal vascularization is more controversial in the literature, since it ranges from 0% in Maizlin et al. in 2014 to close to 45% in Maxwell et al. in 2016 (Figure 1) [10,12].

These differences can be explained by improvements in the sensitivity of ultrasound probes and in the performance of the Doppler mode. The use of ultrasensitive Doppler further improves the detection of intra- and peri-lesional flows and ameliorates identification of the enveloping pattern, which is suggestive of LCTs (Figure 2) [16].

This tool can also help to differentiate predominantly peripheral vascular architecture from radial trans-lesional vascularization, which seems to be more indicative of seminoma (Figure 3) [14].

In patients with very small lesions, a Doppler ultrasound can be tricky, showing either undetectable vascular architecture or, conversely, a high density of blood vessels hampering the visualization of a specific pattern. The supposed potential of malignity of Leydig cells tumors is based on old studies, including big palpable lesions, and such spread is not described for incidentally found small tumors. Nevertheless, a whole-body CT could be recommended for extension screening.

### 3.3. Contrast-Enhanced Ultrasound

The PubMed search using contrast-enhanced ultrasound-CEUS/contrast-enhanced ultrasound yielded 15 references, including four case reports. Through the intravenous injection of microbubbles (Sonovue^®^ or Sonozoid ^®^), (CEUS) allows for qualitative and quantitative analysis of testicular and lesion enhancement. The first studies using CEUS showed that LCTs present early enhancement and that a feeding vessel in the periphery of the tumor could be identified [13,17]. This pattern can also be found using color Doppler (Figure 2b). Lock et al. reported that the CEUS quantitative parameters of time-intensity curves did not distinguish LCTs from seminomas [13]. This was confirmed by Luzurier et al. in their study of 15 LCTs all of which presented early enhancement, followed by wash-in compared with the adjacent pulp (Figure 4) [15].

No significant difference was found between the contrast enhancement of LCTs and that of active malignant testicular tumors. More recently, Drudi et al. found no significant difference between the enhancement pattern of LCTs and of seminomas, even though the quantitative parameters of the time-intensity curves differed significantly (shorter time to peak and higher peak enhancement) [17,18,19]. The authors explained this difference by the greater density and regularity of the vessels of LCTs. Vandaele et al. reported that LCTs presented very rapid and perfectly homogeneous enhancement, whereas the enhancement of seminomas was rapid but with heterogeneous kinetics [20]. These data are, however, subjective and difficult to treat for lesions < 5 mm. As previously stated, color Doppler cannot always correctly assess the vascularization pattern of very small lesions. However, using CEUS, Auer et al. reported improved sensitivity in the detection of vascularization of tumors < 5 mm, without discrimination between seminomas and LCT [21]. Overall, CEUS differentiates active solid tumors from burned-out tumors but does not yet effectively discriminate LCTs from seminomas, the main differential diagnosis. The use of quantitative data extracted from enhancement curves, which requires post-treatment work, is difficult to apply in routine practice. This modality is nonetheless part of a body of evidence based on ultrasound and multimodal MRI data.

### 3.4. Elastography

The PubMed search using the key words Leydig cell tumour/tumor—ultrasound/sonography—elastography—SWE—strain elastography—SE—shear wave yielded 8 references between 2012 and 2021, including 2 case reports.

The use of testicular elastography spread at the end of the 2000s. Most studies use the technique of strain elastography. In 2012, Goddi et al. reported promising results, with a sensitivity/specificity of 87.5%/98.2% in differentiating benign lesions from malignant lesions, though this study included only 2 LCTs among the 103 nodules studied [22]. The first prospective study to include a significant number of LCTs was that of Pozza et al., who studied 106 testicular lesions, 20 of which were LCTs, using strain elastography with a visual scale for stiffness and an analysis of the lesion/healthy parenchyma stiffness ratio [23]. This classification allowed differentiation between tumor lesions and non-tumor lesions, but showed an overlap (visual scale and stiffness ratio) between malignant and benign tumors (including 52% of proven LCTs). More recently, Konstatatou et al. in a retrospective study of 86 lesions, including 12 histologically proven LCTs, showed that the stiffness ratio outperformed the visual scale in differentiating malignant lesions from non-tumor lesions, but no significant difference allowed discrimination between tumor lesions [24]. Shear wave elastography enables quantitative analysis of lesion stiffness. Roy et al. reported quantitative data of median stiffness: 4.55 kPA for healthy testicular parenchyma, 21 kPA for tumor lesions and 30 kPa for fibrosis, but this study excluded LCTs from the analysis [25]. Rocher et al. analyzed 86 testicular lesions, including 28 LCTs, 37 malignant tumors, and 12 burned-out tumors. Shear wave elastography distinguished the LCTs from malignant tumors and burned-out tumors, with an area under the curve of 98% [24,25]. Overall, the convergent literature data show that LCTs, like other tumor lesions, have greater stiffness than the adjacent parenchyma (Figure 5). However, most studies included very small numbers of LCTs, and only Rocher et al. have shown a statistically significant difference between LCTs and malignant lesions (seminomas, non seminomatous germ cell tumors, burned-out lesions) [26].

## 4. Multi Parametric MRI

The PubMed search using the key words Leydig cell tumour/tumor—MRI—yielded 17 references between 1990 and 2021, including two case reports.

With the rising incidence of LCTs and the possibility of conservative surgery or close monitoring, the use of MRI of the scrotum is becoming increasingly widespread. This examination has the advantage that it is reproducible, provides morphological information, and quantifies enhancement. However, given its limited accessibility, MRI is at present a second-line examination.

### 4.1. Morphological T1 and T2 Sequences

As in ultrasound examination, sporadic LCTs are well-delimited solid masses with regular margins. They have a low signal intensity on T2-weighted sequences [27]. In 2015, Manganaro et al. analyzed the MRI appearance of 44 tumors, including 19 LCTs of mean size 6 mm [28]. Two morphological criteria revealed a significant difference between LCTs and seminomas. First, the T2 signal was of low intensity in 89.4% of LCTs vs. 35% of seminomas (*p* = 0.001) and of moderate intensity in 60% of seminomas vs. 5.3% of LCTs (*p* < 0.001). Second, compared with the pulp, 68.4% of LCTs were well-defined vs. 10% of seminomas (*p* < 0.001). The T1 signal did not reveal a statistically significant difference. These results are to be weighed against the MRI findings of El Sanharawi et al., who analyzed 12 benign tumors, including 11 LCTs and 12 malignant tumors that included 10 seminomas [29]. In this study, all LCTs and seminomas presented low-intensity T2 signals and equal homogenous T1 signals, with no differences between these two populations. To date, no case of LCTs with a hemorrhagic, necrotic, or fatty content has been reported in MRI studies (Figure 6).

### 4.2. Functional Sequences: Diffusion-Weighted Imaging—Apparent Diffusion Coefficient

The testicle has a high cellular density, which physiologically restricts apparent diffusion coefficient (ADC) mapping. Tsili et al. reported that the mean ADC in men aged 20–39 years was 1.09 × 10^−3^ mm^2^ s^−1^, with a linear increase in ADC with age [30]. In another study, Tsili et al. showed that analysis of the restriction of diffusion allowed differentiation of malignant lesions from benign lesions, but the scrotal lesions studied were mainly non-tumor lesions, and there were no LCTs [31]. Few studies have specifically analyzed the appearance of LCTs in diffusion-weighted imaging–ADC. In 2018, Manganaro et al. analyzed the weighted diffusion signal b = 1000 s/mm^2^ and reported ADC mapping of 22 malignant lesions (including 20 seminomas of average size 8 mm) and 25 benign tumors (25 LCTs of average size 6 mm) [32]. LCTs restricted diffusion less than seminomas (12.5% vs. 65%), but the difference was not statistically significant, and there was no specific ADC cut-off. The small size at diagnosis of most series of LCTs (6–7 mm in Manganaro et al. and El Sanharawi et al.) limits analysis of diffusion sequences and ADC mapping (Figure 7).

### 4.3. Dynamic Contrast-Enhanced MRI

LCTs are hypervascular lesions and this characteristic is found by Doppler ultrasound and by CEUS. Dynamic contrast-enhanced MRI can be used to analyze quantitative data in order to distinguish LCTs from seminomas and hence to propose testis-sparing surgery. The 2015 prospective study of Manganaro et al. on 19 LCTs showed that scrotal MRI with injection had a sensitivity of 89.47% and a specificity of 95.65% for the diagnosis of LCTs and a precision of 93% in distinguishing between seminoma and LCT [28]. In this study, LCTs presented homogeneous enhancement with rapid and marked wash-in of close to 95%, followed by prolonged wash-out of nearly 79%. These qualitative data accord with those of El Sanharawi et al. who studied the enhancement characteristics of benign testicular lesions (11 LCTs/12 benign tumors) vs. malignant tumors (10 seminomas/12 malignant tumors) vs. burned-out tumors using enhancement curves [28]. The authors found that 10/11 LCTs in this series presented a type 3 curve characterized by sharp uptake, followed by an enhancement decrease: wash-out, a distribution significantly different from malignant or burned-out tumors. The benign tumors (including 11/12 LCTs) of this series presented a shorter time to peak, peak enhancement, and higher transfer parameters (Ktrans and Kep). This study refines the specific enhancement profile of LCTs as it excludes non-tumor lesions, unlike the study of Watanabe et al. [33]. In 2018, Manganaro et al. confirmed these data by a prospective study of 47 testicular nodules: 22 malignant tumors (including 20 seminomas) and 25 benign tumors (including 24 LCTs). The LCTs presented higher wash-in parameters (peak enhancement, time to peak), higher transfer parameters (Ktrans, Ker), and a shorter time to peak. The authors put forward cut-offs for these quantitative parameters for the differential diagnosis of seminoma for which the enhancement is less marked and less rapid than that of LCTs (Figure 8).

## 5. Leydig Cell Hyperplasia/Tumor Associated with Klinefelter Syndrome

Klinefelter syndrome (KS) or 47,XXY (gonosomal aneuploidy) affects 0.1–0.2% of the general population, but the incidence rises to 3.1% in the infertile population [34]. Patients with KS are therefore regularly encountered during an infertility work-up. The ultrasound appearance of the testicles of KS patients is characteristic, and it should alert the radiologist. Rocher et al. recorded the scrotal ultrasound exams of 67 infertile men with KS and 66 non-KS non-obstructive azoospermic men, with histological comparison [9]. The KS patients had severe testicular hypotrophy (mean testis volume 2 mL), and a coarse or micronodular (nodules < 3 mm) echotexture versus a normal/striated echotexture in the infertile non-KS patients. The KS patients presented increased bilateral and symmetrical microlithiasis, unlike the infertile non-KS patients (28% vs. 4.5%). Histology confirmed that the nodules detected in KS patients were benign LCTs associated with Leydig cell hyperplasia surrounded by Sertoli cell involution and seminiferous tubule degeneration. Some patients present with mosaic Klinefelter syndrome: an extra X chromosome in some cells with generally fewer symptoms. Our experience with mosaic Klinefelter syndrome is based on nine patients included in a series of 67 patients with KS syndrome: there were quite similar in testis volume (2 ± 0.9 mL versus 2.4 ± 1.44 mL per testis), testosterone level (10.1 ± 5.2 nmol/mL versus 11.1 ± 4.85 mL), and FSH level (33.7 ± 16.2 UI/L versus 28.9 ± 11.5 UI/L) [9]. Severe testicular hypotrophy with hypervascularized bilateral micronodules should prompt the radiologist to search for KS. Radical orchiectomy risks worsening the patient’s testosterone deprivation and should be avoided in this situation. Even though KS patients are supposed to be at risk of testicular germ cell tumors, it appears that most nodular anomalies of the testicle found in these patients are linked to Leydig cell hyperplasia/tumors (Figure 9). However, the radiologist should keep in mind that KS patients are at risk of extra gonadic germ cell tumors, especially located in the mediastinum [35].

## 6. Conclusions

Improvement in characterization of sporadic Leydig cell tumors using B Mode US, Color/ultrasensitive Doppler, Contrast-enhanced Ultrasound, Elastography, and multiparametric MRI leads to a better management of such lesions: testis-sparing surgery or even follow up can now be suggested based on advanced imaging. The radiologist must recognize the specific echostructure of Klinefelter syndrome patients due to Leydig cell hyperplasia or small LCT to avoid useless and potentially harmful surgery.

## Figures and Tables

**Figure 1 cancers-14-03652-f001:**
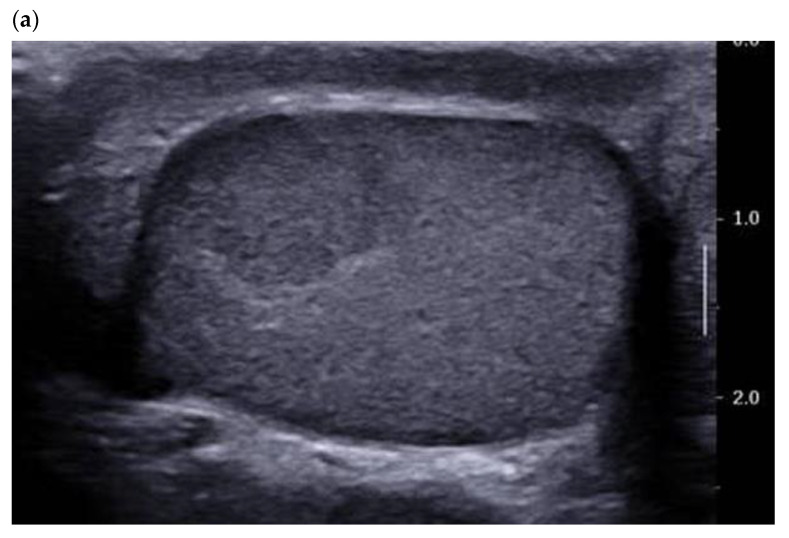
28-year-old patient presenting with left testicular pain (significant left varicocele during the exam). Typical LCT 12 × 9 mm on B mode us (**a**) and color Doppler (**b**) discovered on the right testis. Well-defined lobulated solid lesion moderately hypoechoic and homogeneous echo structure with normal adjacent pulp and absence of microlithiasis. The lesion is hyper vascularized with a mixed peripheral and internal pattern. (**c**) From left to right: Macroscopic view of the patient’s LCT after enucleation. The typical “golden brown” color of the lesion often allows the surgeon and the pathologist to confirm the diagnosis during surgery. HE × 30 Hematein–Eosin coloration showing a high cellular density with no necrosis. HE × 40 with an endothelial cell marker anti CD-31, showing a rich vascularization of the tumor. Courtesy of Pr S. Ferlicot, Department of Anatomo-pathology, Bicêtre Hospital.

**Figure 2 cancers-14-03652-f002:**
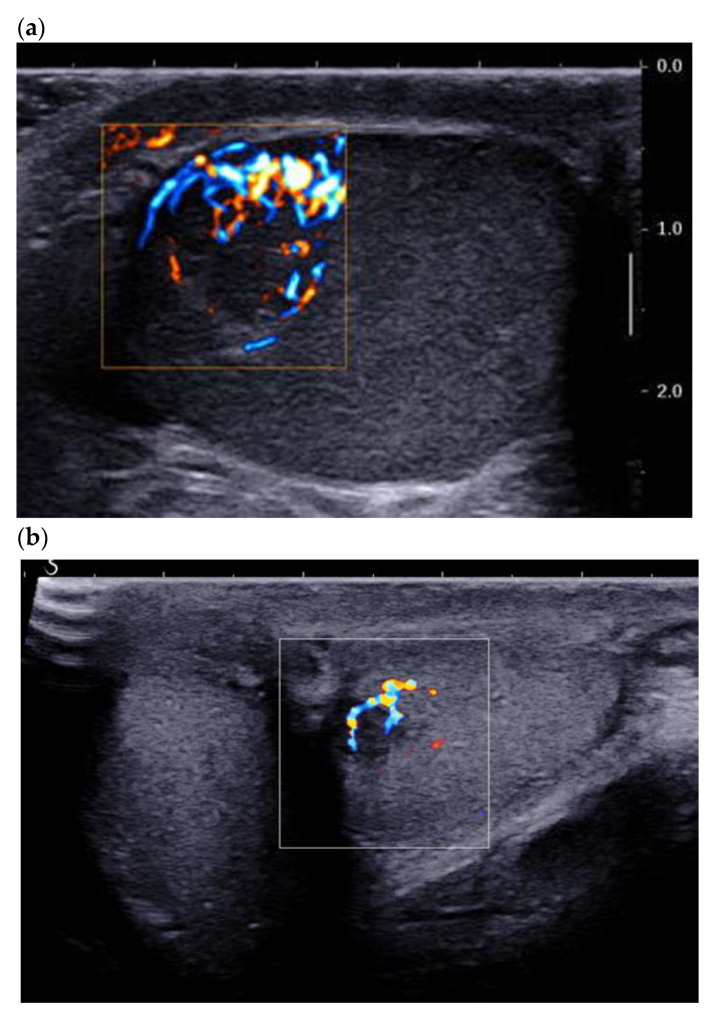
(**a**) Same lesion as Figure 1 with ultrasensitive Doppler allowing better characterization of corbelling vascular architecture. (**b**) Small LCT of 7 mm, marked hyper vascularization compared with adjacent pulp, with main feeding vessel pattern.

**Figure 3 cancers-14-03652-f003:**
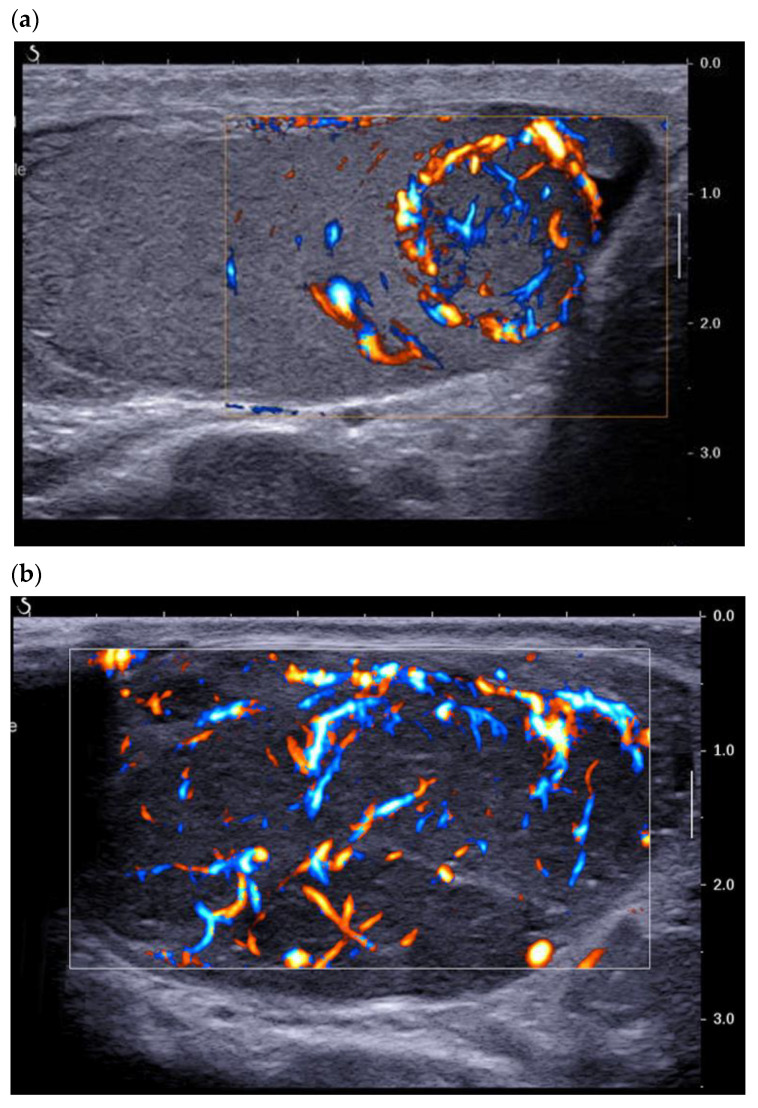
(**a**) Typical LCT 16 × 14 mm with mixed corbelling and central vascularization pattern in a 42-year-old patient presenting for infertility work up. The large size of the lesion helps to highlight this typical pattern. (**b**) 31-year-old patient presenting for right testicular mass: 38 mm typical seminoma with a radial trans-lesional « anarchic » vascularization pattern.

**Figure 4 cancers-14-03652-f004:**
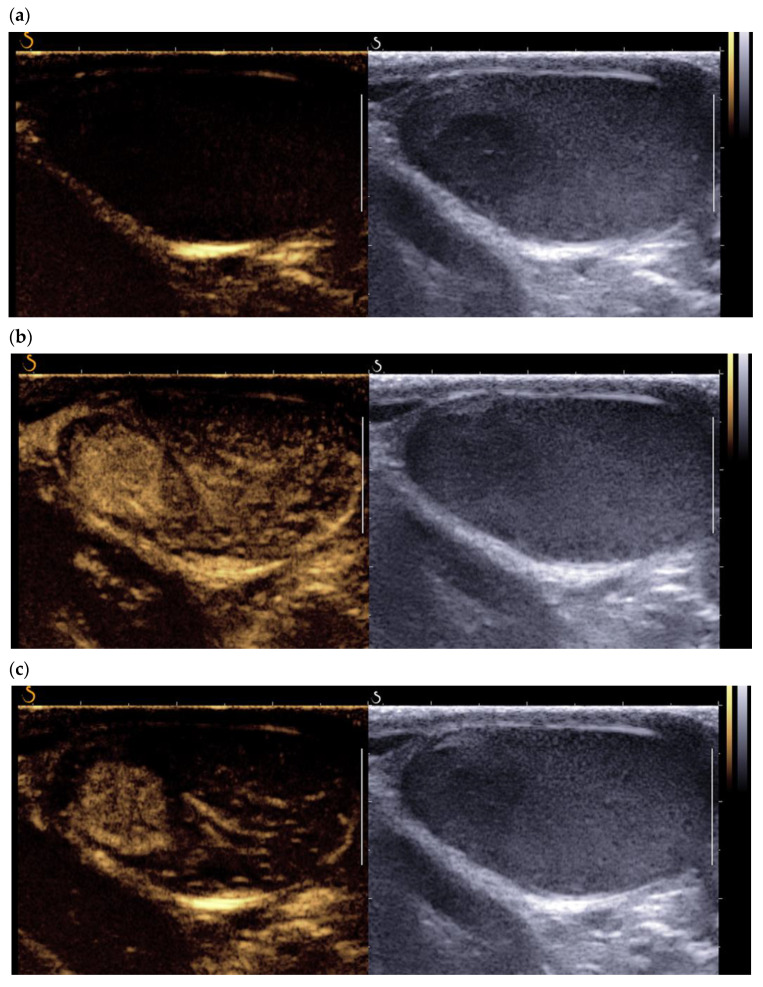
Contrast-enhanced ultrasound using Sonovue^®^ of a typical LCT (**a**) t = 0 s (**b**) t = 30 s (**c**) t = 60 s found in a 45-year-old patient addressed for infertility work up.

**Figure 5 cancers-14-03652-f005:**
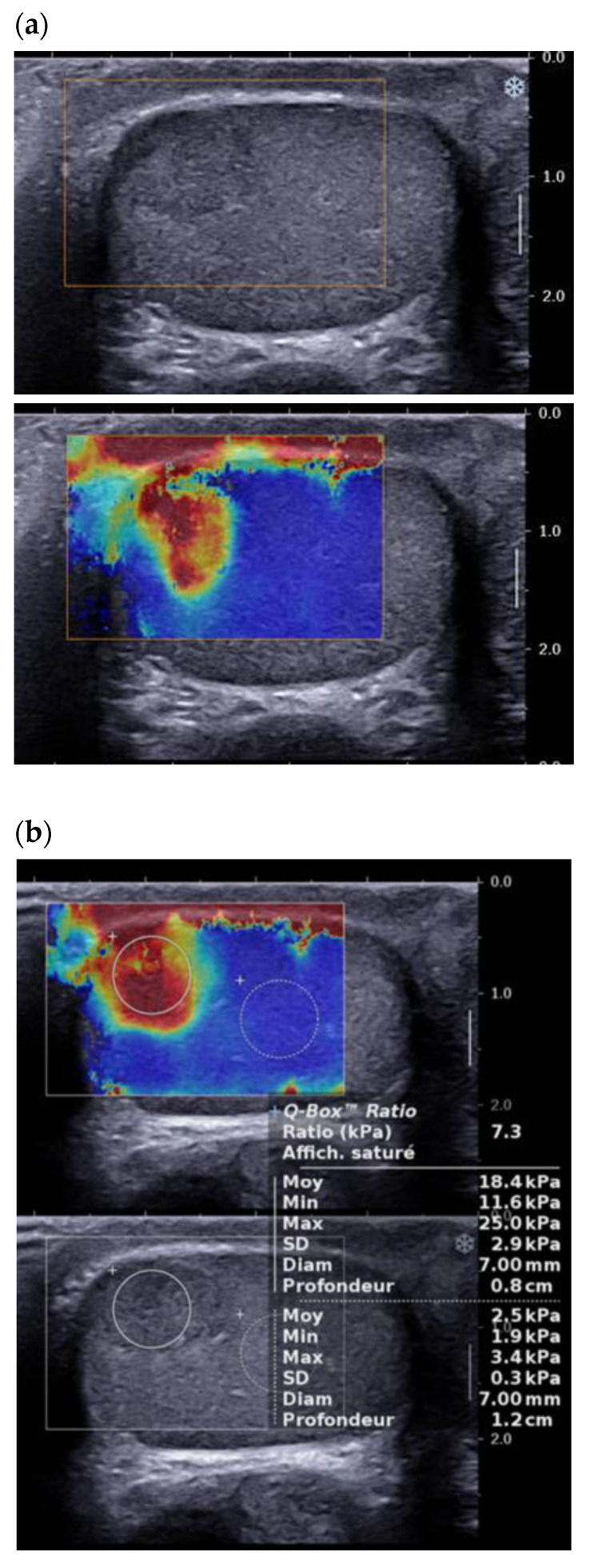
Same patient as Figure 1: Shear wave elastography of typical LCT, qualitative (**a**) and quantitative (**b**) evaluation of stiffness.

**Figure 6 cancers-14-03652-f006:**
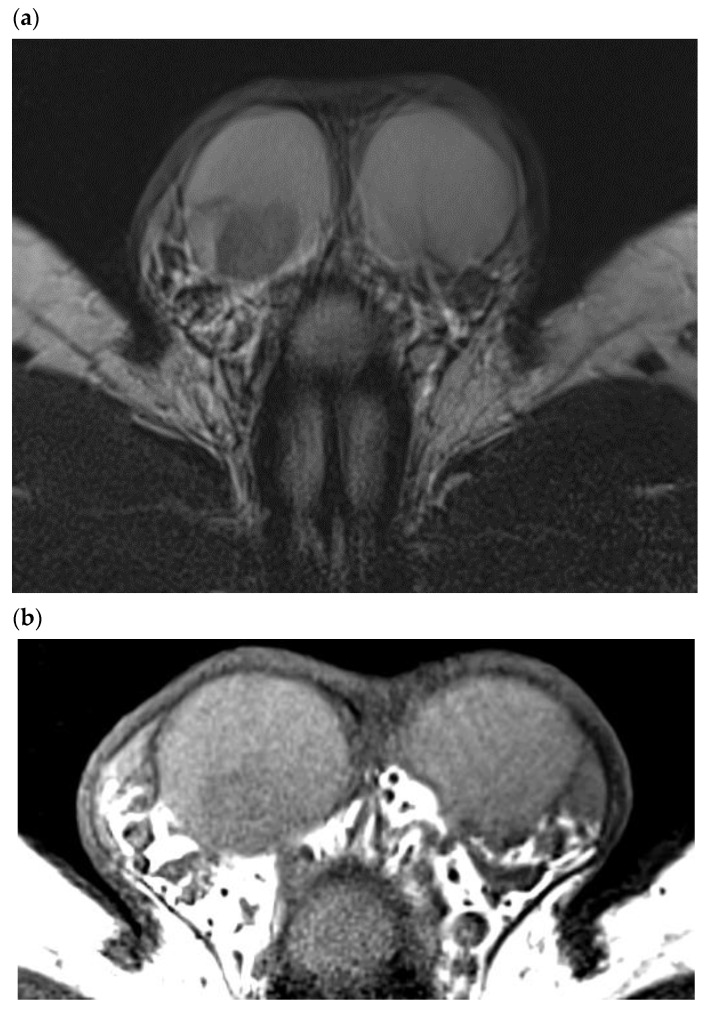
44-year-old patient, infertility work up, right testicular tumor on first line sonography. MRI (3T) aspect of a typical LCT. (**a**) T2 weighted sequence, Axial: LCT of the right testis appears as a solid well-circumscribed lesion with a sharp demarcation with the rest of the testis and a homogeneous marked hypo T2 signal. Note the absence of alteration of the signal of the testicular pulp apart from the LCT. (**b**) T1 weighted sequence, Axial: LCT has a homogeneous hypo T1 signal, with no cystic or hemorrhagic component.

**Figure 7 cancers-14-03652-f007:**
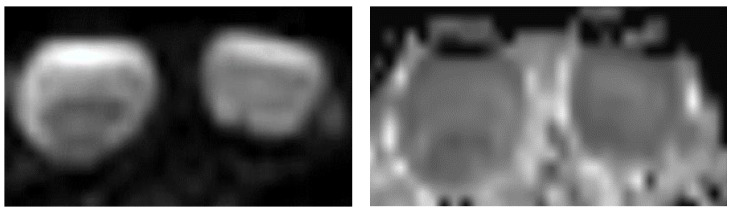
Same patient as Figure 6. Diffusion-weighted sequence b800 and ADC map.

**Figure 8 cancers-14-03652-f008:**
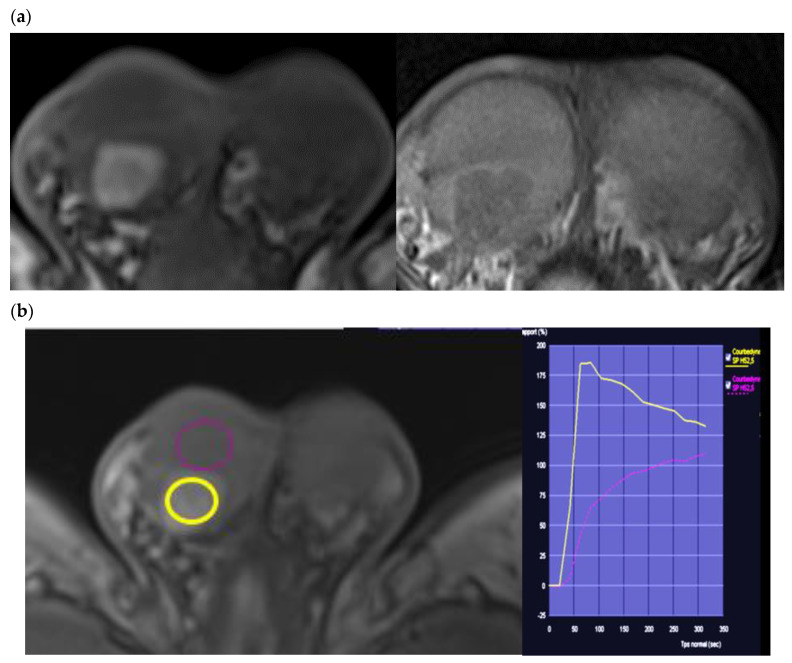
(**a**) Same patient as Figure 6. T1 weighted sequence with fat suppression and after dynamic gadolinium contrast media injection. Fast and intense uptake of contrast media before the rest of the testis and wash-out on the late phase (6 min post contrast media injection). (**b**) Post treatment using Syngovia^®^ workstation: Dynamic enhancement curve shows a shorter and a higher time to peak, followed by a prolonged wash-out and a higher area under curve compared to normal pulp.

**Figure 9 cancers-14-03652-f009:**
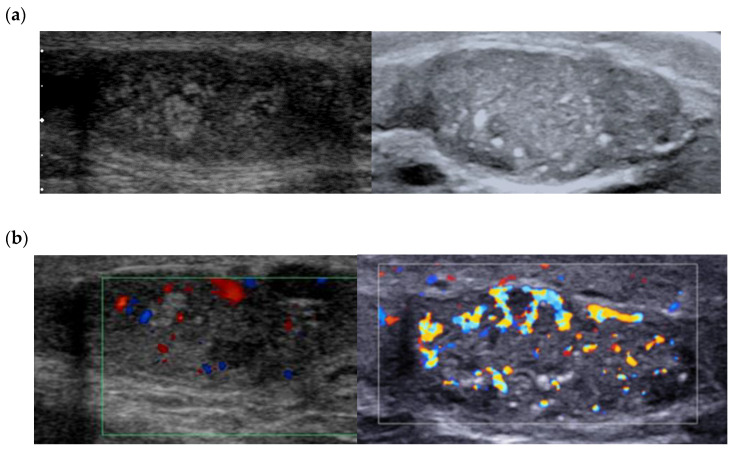
26-year-old patient addressed for infertility work up/Klinefelter’s syndrome follow up. Leydig cell hyperplasia in a patient with Klinefelter’s syndrome. (**a**) Two patients with major testicular hypotrophy (2 mL), with coarse echostructure and micronodules mostly hyperechoic. (**b**) Color Doppler ultrasound showing diffuse hypervascularization of the testicle, which contrasts with the relatively hypovascularized aspect of undescended testicles.

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
