# Peer review of "Leydig Cell Tumors of the Testis: An Update of the Imaging Characteristics of a Not So Rare Lesion"

_cancers, 2022, doi:10.3390/cancers14153652_

Round 1

Reviewer 1 Report

Maxwell et al. reviewed scientific literature focusing on the imaging characteristics of Leydig cell tumors of testis. It is an interesting paper, very well written. 

It could be interesting for the readership of Cancers Journal, to improve the Introduction paragraph (i.e. authors could summarize better the molecular features of Leydig cell tumors, the molecular diagnostic methods etc.)

Reviewer 2 Report

Dear Authors,

This is an interesting review.

Here are my suggestions.

1.    Title – No period at the end of the title should be used

2.    Abstract – line 18 – “Testicular tumors characterization is a relatively new topic.” The topic is not new; some new data on their approach (especially concerning the imaging approach, immunohistochemistry markers, and management) are released during the recent years. Please re-structure the paragraph.

3.    Abstract – “Most of the literature focused on germ cell tumors and few data concerned stromal tumors especially Leydig cell tumors (LCT).” You should introduce the tumor first (which is the subject of the paper), then the differential diagnostic.

4.    Abstract – Conclusion – I suggest you introduce a paragraph/conclusion based on your review/data/analysis

5.    Introduction – I suggest you start by introducing the classification of testes tumors, then you introduce the stromal group, then LCT with prevalence/epidemiology data. Then you focus on data concerning LCT (different aspects). As the title which contains a negation, you tend to say what is not, rather than what is, thus it is difficult to follow the information you present.

6.    Figure 1. Please provide the dimensions of the lesions, and, also, some clinical data of the patients (like 50-year old patient....)

7.    Line 88. “These results accord with those reported by Maizlin et al. and Lock et al.1” Is this reference no. 1 (you cite two authors and none of them is included in reference no 1)?

8.    Figures – If you have any correlations data concerning the histological report and the evolution of the patients who were diagnosed based on your procedures, it adds a great practical value to the interesting captures

9.    Line 175 – et al. – please use Italics and period after ”al”

10. Do you have any data on Klinefelter’s syndrome with mosaic? How about other genetic conditions with a higher risk of LCT? Also, some hormonal and molecular/immunohistochemistry correlations/observations with imaging procedures should be useful.

Thank you,

Best regards,
